# A 'resource allocator' for transcription based on a highly fragmented T7 RNA polymerase

Thomas H Segall-Shapiro[1], Adam J Meyer[2], Andrew D Ellington[2], Eduardo D Sontag[3] & Christopher A Voigt[1,*]

## Abstract

Synthetic genetic systems share resources with the host, including machinery for transcription and translation. Phage RNA polymerases (RNAPs) decouple transcription from the host and generate high expression. However, they can exhibit toxicity and lack accessory proteins (σ factors and activators) that enable switching between different promoters and modulation of activity. Here, we show that T7 RNAP (883 amino acids) can be divided into four fragments that have to be co-expressed to function. The DNA-binding loop is encoded in a C-terminal 285-aa 'σ fragment', and fragments with different specificity can direct the remaining 601-aa 'core fragment' to different promoters. Using these parts, we have built a resource allocator that sets the core fragment concentration, which is then shared by multiple σ fragments. Adjusting the concentration of the core fragment sets the maximum transcriptional capacity available to a synthetic system. Further, positive and negative regulation is implemented using a 67-aa N-terminal 'α fragment' and a null (inactivated) σ fragment, respectively. The α fragment can be fused to recombinant proteins to make promoters responsive to their levels. These parts provide a toolbox to allocate transcriptional resources via different schemes, which we demonstrate by building a system which adjusts promoter activity to compensate for the difference in copy number of two plasmids.

**Keywords** genetic circuit; resource allocation; split protein; synthetic biology; T7 RNA polymerase

**Subject Categories** Synthetic Biology & Biotechnology; Methods & Resources

**Mol Syst Biol. (2014) 10: 742**

See also: **DL Shis & MR Bennett** (July 2014)

## Introduction

Cells must control the production of RNA polymerase (RNAP) and ribosomes to balance their biosynthetic cost with the needs of cell growth and maintenance (Warner, 1999). As such, RNAP and ribosome synthesis is under stringent regulatory control, both to coordinate their levels with respect to cellular and environmental cues for growth (Nierlich, 1968; Hayward *et al*, 1973; Iwakura & Ishihama, 1975; Bedwell & Nomura, 1986; Bremer & Dennis, 2008; Schaechter *et al*, 1958; Lempiäinen & Shore, 2009; Gausing, 1977; Schneider *et al*, 2003) and to balance the expression of their components for proper assembly into functional machines (Warner, 1999; Ishihama, 1981; Nierhaus, 1991; Fatica & Tollervey, 2002). This sets a resource budget that must be shared in the transcription of approximately 4,000 genes and translation of $\sim 10^6$ nucleotides of mRNA in *E. coli* (Bremer & Dennis, 1996). The budget is not large; on average, there are 2,000 RNAP and 10,000 ribosomes per cell (Ishihama *et al*, 1976; Bremer & Dennis, 1996; Ishihama, 2000). Mathematical models often assume these budgets to be constant (Shea & Ackers, 1985; Gardner *et al*, 2000; Elowitz & Leibler, 2000), but the numbers can vary significantly in different growth phases and nutrient conditions, ranging from 1,500 to 11,400 RNAPs and 6,800 to 72,000 ribosomes per cell (Bremer & Dennis, 1996; Klumpp & Hwa, 2008). The fluctuations in resources can lead to global changes in expression levels and promoter activities (Keren *et al*, 2013; De Vos *et al*, 2011).

This poses a problem when a synthetic genetic system is introduced. When it relies on the transcription and translation machinery of the host, it becomes implicitly embedded in their regulation, making it sensitive to changes that occur during cell growth and function. As a result, the system can be fragile because the strengths of its component parts (promoters and ribosome binding sites) will vary with the resource budgets (Moser *et al*, 2012; Arkin & Fletcher, 2006; Kittleson *et al*, 2012). For example, changes in the RNAP concentration can impact the expression from constitutive promoters by fivefold (Bremer & Dennis, 1996; Liang *et al*, 1999; Klumpp *et al*, 2009; Liang *et al*, 2000; Klumpp & Hwa, 2008). These changes can reduce the performance of a system that requires precise balances in expression levels (Temme *et al*, 2012b; Moser *et al*, 2012; Moon *et al*, 2012). This has emerged as a particular problem in obtaining reliable expression levels and gene circuit performance during industrial scale-up, where each phase is associated with different growth and media conditions (Moser *et al*, 2012).

1  Department of Biological Engineering, Synthetic Biology Center, Massachusetts Institute of Technology, Cambridge, MA, USA
2  Institute for Cellular and Molecular Biology, University of Texas at Austin, Austin, TX, USA
3  Department of Mathematics, Rutgers University, Piscataway, NJ, USA
   *Corresponding author. Tel: +1 617 324 4851; E-mail: cavoigt@gmail.com

Another problem is that synthetic systems often place high demands on host transcription and translation resources and this can have global consequences in maintaining growth and responding to stress (Hoffmann & Rinas, 2004; Birnbaum & Bailey, 1991). Proteins and pathways expressed at very high levels place a burden on cells that can reach up to 30% of total cellular proteins and utilize 50% of translation capacity (Dong *et al*, 1995; Scott *et al*, 2010; Carrera *et al*, 2011). The competition with native genes can cause a decrease in their expression and a reduction or cessation of growth (Dong *et al*, 1995; Scott *et al*, 2010; Carrera *et al*, 2011; Tabor *et al*, 2008). In addition, because of the small numbers of RNAP and ribosomes, the expression of recombinant genes can become coupled, where a high level of expression of one gene titrates a resource and reduces the expression of another gene. In the context of synthetic signaling networks, this has been referred to as 'retroactivity', where downstream targets can impart a load on the upstream signaling pathway (Jiang, *et al*, 2011; Jayanthi *et al*, 2013; Del Vecchio *et al*, 2008; Del Vecchio & Murray, 2014).

These challenges were recognized early in biotechnology and a partial solution emerged by using the RNAP from T7 phage to decouple transcription from the host machinery (Chamberlin *et al*, 1970; Studier & Moffatt, 1986; Alexander *et al*, 1992). Heterologous T7 RNAP was patented in 1984 (Studier *et al*, 1990) and since then has been the basis for expression systems across many organisms (Elroy-Stein & Moss, 1990; Brunschwig & Darzins, 1992; McBride *et al*, 1994; Conrad *et al*, 1996). An advantage cited for this system was that it could achieve high expression levels by adding an inhibitor of *E. coli* RNAP, thus directing metabolic resources to recombinant protein production (Tabor & Richardson, 1985). However, there are also some challenges with using T7 RNAP. While the polymerase itself is not toxic, when it is combined with a strong promoter, it can cause severe growth defects. The origin of this toxicity is not clear, but it could be related to the rate of transcription of T7 RNAP, which is eightfold faster than *E. coli* RNAP and could expose naked mRNA (Iost *et al*, 1992; Miroux & Walker, 1996). Toxicity can be ameliorated by introducing a mutation near the active site and by selecting parts to lower polymerase expression (Temme *et al*, 2012a,b). Beyond the RNAP from T7, many polymerases have been identified from different phage and directed evolution experiments have yielded variants that recognize different promoter sequences (Temme *et al*, 2012a; Ellefson *et al*, 2013; Carlson *et al*, 2014).

Phage polymerases are central to our organization of larger genetic systems (Temme *et al*, 2012a,b; Smanski *et al*, 2014). We separate the regulation of a system (on a plasmid we refer to as the 'controller') from those genes encoding pathways or cellular functions ('actuators') (Fig 1A). The controller contains synthetic sensors and circuits, whose outputs are phage polymerases specific to the activation of the actuators. This organization has several practical advantages. First, it avoids evolutionary pressure when manipulating the actuators because the promoters are tightly off in the absence of phage polymerase. Thus, they can be carried in an inactive state until the controller is introduced into the cell. Actuators often require many genes and assembled parts, making re-verification of their sequence expensive. Second, it allows the regulation of the actuators to be changed quickly. Controllers can be swapped to change the conditions and dynamics of expression, so long as they produce the same dynamic range in output polymerase expression. In the same way, the controllers can also be

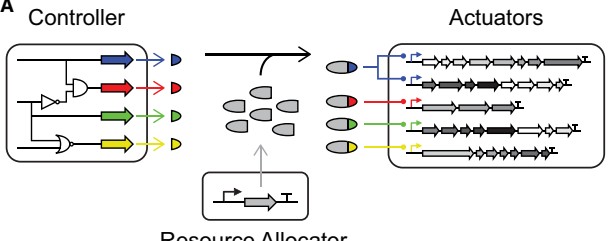

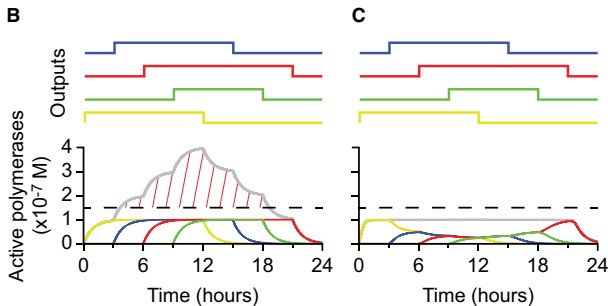

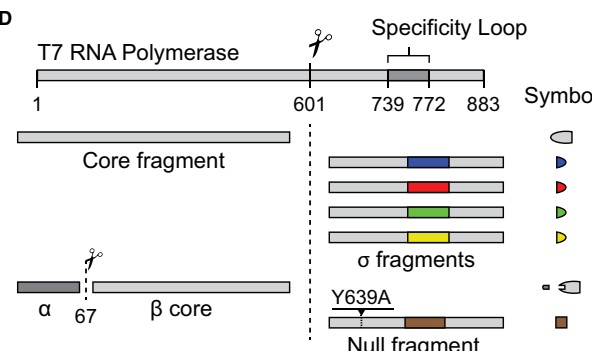

**Figure 1. The resource allocator.**

A  Complex synthetic genetic systems are broken down into three modules. The core fragment of RNAP is expressed from the resource allocator. Each output from the controller results in the expression of a different σ fragment (colored half-circles), which share the core fragment and turn on different actuators.

B  Dynamic simulations of resource allocation are shown, where the outputs from the controller are turned on and off at different times (colored lines) (Supplementary information Section IV.A). A hypothetical toxicity threshold is shown with the dashed horizontal line. When the outputs of the controller are complete RNAPs, their sum crosses the threshold (gray line and red hash).

C  With resource allocation, the outputs of the controller are σ fragments that must share the core fragment, thus ensuring that their sum transcriptional activity does not cross the threshold.

D  The complete toolbox of phage RNAP fragments is shown.

characterized independently using surrogate fluorescent reporters prior to being combined with the actuators.

With these large and complex synthetic systems, problems can arise as the host is subjected to significant perturbation and load. Simultaneously activating a number of actuators requires expressing multiple polymerases that might collectively cross the threshold for toxicity (Fig 1B). While lowering expression rates throughout the

system could avoid toxicity, it would needlessly constrain expression when only one actuator is active. To address this issue, we aimed to create an allocation system that allows independently setting the total desired polymerase activity and allocating this resource to the various actuators as needed. With this organization, a single actuator can be expressed to full strength, but expression of multiple actuators is attenuated to avoid overexpression (Fig 1C). In effect, we are proposing to add another layer to the organization of genetic designs, where a separate 'resource allocator' is responsible for the maintenance of a desired level of orthogonal transcriptional machinery (Fig 1A).

Prokaryotes solve the problem of partitioning a budget of RNAP to different cellular processes through the action of σ factors, which bind to the core RNAP (α2, β, β′, and ω subunits) and direct it to promoter sequences (Gruber & Gross, 2003; El-Samad *et al*, 2005). Core RNAP itself only has the ability to non-specifically bind to DNA, whereas the σ factor contains the DNA recognition domains for the −35 and −10 regions of promoters. Different σ factors bind to distinct promoter recognition sequences. In *E. coli*, there is one 'housekeeping' σ factor ($\sigma^{70}$) that is expressed at a constant level of 500–700 molecules/cell, independent of growth phase or stress, and 6 alternate σ factors that control various stress responses (e.g., heat shock) and cellular functions (e.g., flagella assembly) (Jishage *et al*, 1996). σ factors can range in size; $\sigma^{70}$ is 613 amino acids and the average alternative σ is ~200 amino acids (Burton *et al*, 1981; Staroń *et al*, 2009; Rhodius *et al*, 2013). These alternative σs can be embedded in complex regulatory networks that implement signal integration and feedback regulation that mimics engineering control architectures (Lange & Hengge-Aronis, 1994; Hengge-Aronis, 2002; Kurata *et al*, 2001). In this way, the level of core RNAP dictates the total transcriptional potential in the cell, while the relative levels of σ factors determine how this resource is allocated between growth and stress resistance (Nyström, 2004; Maharjan *et al*, 2013). Bacteria with more diverse lifestyles can have significantly more σ factors, for example, *Streptomyces* and *Bacteroides* species can have greater than 50 (Lange & Hengge-Aronis, 1994; Hengge-Aronis, 2002; Kurata *et al*, 2001). All of these σs compete to bind to the core RNAP (Ishihama, 2000; Gruber & Gross, 2003).

In this manuscript, we have created an analogous system by fragmenting T7 RNAP. We used a transposon method to identify five regions where the polymerase can be bisected and retain function. One of these splits produces a 285 amino acid fragment that we refer to as the 'σ fragment' because it contains the region that binds to the promoter (Fig 1D). We find that variants of this fragment with different promoter specificities can bind to the remaining 'core fragment' and direct it to different promoters. The expression level of the core fragment dictates the maximum number of active polymerases. The outputs of the controller are different σ fragments, which are used to turn on different actuators. If the pool of core fragments is saturated by σ fragments, the total number of active polymerases in the system will remain constant regardless of the levels of σ fragments being produced (Fig 1C). In this way, a desired transcriptional load can be specified and then dynamically allocated to different actuators as the conditions require. Negative regulators can be built by creating null σ fragments that titrate the core fragment but do not support transcription. Additionally, the core fragment can be positively regulated using the N-terminal bisection point to separate an 'α fragment' that is required for activity. These regulators

could be used to implement feedback loops that control the amount of active RNAP complexes under different conditions or the dynamics of signal progression from the controller to the actuators.

## Results

### Bisection mapping of T7 RNA polymerase

Our first objective was to identify all of the places T7 RNAP could be split to yield two fragments that can be co-expressed to produce a functional protein. To do this, we developed a transposase-based method that uses a novel transposon to split proteins, which we refer to as a 'splitposon'. Previous methods have been published to generate libraries of split proteins or domain insertions that are based on incremental truncation (Ostermeier *et al*, 1999; Paschon & Ostermeier, 2004), multiplex inverse PCR (Kanwar *et al*, 2013), DNAse cleavage (Guntas & Ostermeier, 2004; Chen *et al*, 2009), and transposon insertion (Segall-Shapiro *et al*, 2011; Mahdavi *et al*, 2013). The transposon-based approaches are able to generate large libraries and do not require sensitive DNAse steps, but they leave ~10 added amino acids at the split site. To improve on this approach, the splitposon is a Mu transposon in which one terminal transposon recognition end is altered to contain a non-disruptive ribosome binding site (RBS) and start codon (Fig 2A). We further modified the transposon to add the remaining necessary regulation to divide a protein into two fragments (stop codon—$P_{Tac}$ IPTG-inducible system—RBS—start codon). The MuA transposase efficiently yields random insertions of the splitposon throughout a DNA molecule, producing a library of split proteins flanked by just three additional amino acids for in-frame insertions (Supplementary Fig S1).

With the splitposon, a bisection library for any protein can be generated in two steps (Fig 2A). First, the splitposon is transposed *in vitro* into a plasmid containing the DNA within which bisections are desired (e.g., a gene or segment of a gene). Second, the target region is digested from the plasmid backbone and size selected for fragments containing an inserted transposon. These fragments are ligated into an expression plasmid containing an upstream inducible promoter. The final library will contain only plasmids with a single transposon insertion in the region of interest and can be induced and screened for function.

The splitposon method was applied to generate a library of bisections of a variant of T7 RNAP (T7* RNAP). This gene contains the R632S mutant, which reduces host toxicity (Temme *et al*, 2012a). To avoid trivial truncations of the termini, we directed transposon insertions to the region of the gene corresponding to amino acids 41 through 876 of the polymerase. Both fragments are induced with IPTG from $P_{Tac}$. The library was co-transformed with a screening plasmid that contains a T7 RNAP dependent promoter and red fluorescent protein (RFP) (Temme *et al*, 2012a), and 384 clones were picked by eye from agar plates, re-assayed in liquid media, and the best 192 sequenced. From these, 36 unique in-frame split sites were identified (Fig 2B). The split sites cluster into five distinct seams that correspond to six potential fragments if they were all implemented simultaneously. The seam around position 179 corresponds to a previously identified split site that yields a functional T7 RNAP (Ikeda & Richardson, 1987a,b; Muller *et al*, 1988; Shis & Bennett, 2013).

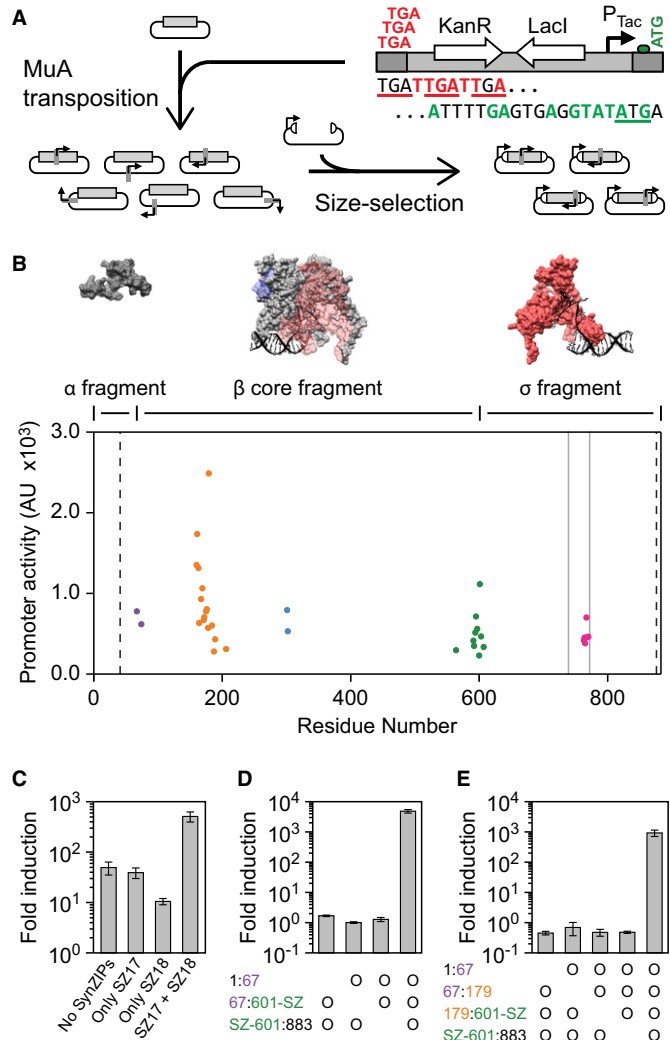

**Figure 2.  Bisection mapping of T7* RNAP.**

A    The splitposon is based on a modified mini-Mu transposon mutated to contain staggered stop codons in one recognition end (red) and an RBS & start codon in the other (green). An internal inducible system (LacI and $P_{Tac}$) has been added. Bisection mapping includes two cloning steps. First, the splitposon is transposed randomly into a gene using MuA transposase. Second, the library is size selected for inserts that contain one transposon insertion and cloned into an expression plasmid.

B    Each point represents a unique in-frame split location in T7* RNAP, where the residue number is the final residue in the N-terminal fragment. The promoter activity is the mean $P_{T7}$ activity for all recovered clones at each split point, from four independent assays (10 μM IPTG induction). Bisection points are clustered into five 'seams', which are color-coded. The vertical dashed lines show the region where bisections were allowed in the library, and the gray vertical lines show the location of the promoter specificity loop. Surface models are shown for the three fragments used for the resource allocator (PDB:1QLN (Cheetham & Steitz, 1999), visualized using UCSF Chimera (Pettersen *et al*, 2004)). The model for the β core fragment shows the position of the α and σ fragments in transparent blue and red, respectively. More views of the surface model are shown in Supplementary Fig S4.

C    The fragments created from splitting T7 RNAP at residue 601 were assayed with and without SynZIP domains at low expression levels (4 μM IPTG). When SynZIP 17 (SZ17) is fused to the N-terminal fragment and SynZIP 18 (SZ18) is fused to the C-terminal fragment, a large increase in the induction of $P_{T7}$ is observed. Fold induction is calculated as the $P_{T7}$ promoter activity in induced cells divided by the promoter activity of cells that contain the reporter plasmid but no polymerase fragments.

D    Data are shown for the expression of the three fragments corresponding to the α fragment (1:67), β core fragment (67:601-SZ), and σ fragment (SZ-601:883). An 'o' indicates the presence of a fragment in an operon that is expressed with 100 μM IPTG.

E    Data are shown for the induction of four fragments, as in (D), with an additional split of the β core fragment at residue 179.

Data information: For the graphs in (C–E), the mean is shown for three independent assays performed on different days, with error bars showing standard deviation.

Source data are available online for this figure.

the single fragments. RNAP activity (4,000-fold induction) is only detected when all three fragments are expressed and there is no activity in the absence of any fragment (Fig 2D). We also tested a four fragment version, which includes a split at position 179 (Fig 2E). The expression of these four fragments yields active RNAP (900-fold induction), and there is no detectible activity if any of the fragments are not expressed.

While the four and three-piece polymerases do lead to a reduction in cell growth when expressed at high levels, this effect is more pronounced when expressing the full-length protein (Supplementary Fig S12). Splitting the polymerase into five or six fragments was not attempted due to the attenuation of activity and growth impact of high expression with four fragments.

## Construction of 'σ fragments' with different promoter specificities

The C-terminal fragment generated by the split site at residue 601 (601–883) contains the DNA-binding loop that determines promoter specificity (Cheetham *et al*, 1999). Thus, we refer to this as the 'σ fragment' as it functions analogously to σ factors that bind to *E. coli* RNAP and is approximately the same size. Following this analogy, the 601 amino acid N-terminal fragment is referred to as the 'core fragment'. Note that this fragment is much smaller than the α2/β/

## Division of T7 RNAP into multiple fragments

All of the discovered split seams occur in surface-exposed regions of the T7* RNAP, and the largest seam corresponds to a large surface-exposed loop known as the 'Flap' in the 3-dimensional structure (Supplementary Fig S3) (Tahirov *et al*, 2002). This implies that additional functional domains can be inserted at these positions. We hypothesized that the addition of protein–protein interaction domains could improve the affinity of the fragments. To this end, two leucine zipper domains that bind in an antiparallel orientation were chosen from the SynZIP toolbox (variants 17 and 18) (Reinke *et al*, 2010; Thompson *et al*, 2012). Addition of either SynZIP at the 601 split site with a short flexible linker is tolerated by the split polymerase, and adding both is beneficial and improves activity by greater than tenfold at low expression levels (Fig 2C).

The outcome of the bisection mapping experiment also implied that it might be possible to divide T7* RNAP into more than two fragments. First, the protein was divided into three fragments based on the split points at residues 67 and 601, including the added SynZIPs at the 601 split. These three fragments were expressed as a single inducible operon and compared to versions lacking each of

β'/ω subunits of *E. coli* RNAP (329/1342/1407/91 amino acids) and they assemble into a very different 3-dimensional structure (Sousa *et al*, 1993; Vassylyev *et al*, 2002; Opalka *et al*, 2010).

A simple resource allocator was built based on the core and σ fragments (Fig 3A), retaining the amino acids added by the splitposon method and the SynZIP 18 domain on the σ fragment. The core fragment is expressed from the constitutive promoter $P_{J23105}$, tuned to a low level such that expressing full-length polymerase in its place is not toxic. The σ fragment is expressed at varying levels using an IPTG-inducible $P_{Tac}$ promoter. Polymerase activity is measured using $P_{T7}$ driving green fluorescent protein (GFP) (Materials and Methods). The σ fragment, core fragment, and reporter are carried on three separate plasmids (p15A*, BAC, pSC101) to mimic the controller, resource allocator, and actuator organization (Fig 1A).

For the resource allocation scheme to function correctly, σ fragments need to saturate the core fragment, causing total RNAP activity to plateau above a certain total concentration of σ fragments. The maximum level of polymerase activity is then set by the concentration of the core fragment, independent of changes in σ fragment expression (Fig 1C). Core fragment expression, and thus overall maximum functional polymerase expression, can be modulated by selecting constitutive promoters and RBSs of different strengths. This saturation behavior is observed when the core fragment is fused to the SynZIP 17 domain (Fig 3B, red points). The RNAP activity saturates approximately fourfold below that obtained with the expression of full-length T7* RNAP in place of the core fragment, which does not change as a function of σ fragment expression (green points). Since the full-length T7* RNAP is expressed at a level equivalent to the core fragment, this indicates that the split polymerase with SynZIPs has about one quarter the activity of full-length T7* RNAP. Without the SynZIP domain on the core fragment, the σ fragment binds with much lower affinity and does not reach saturation even at high levels of expression (blue points). Because the desired saturation of the core fragment is obtained only with the SynZIPs, they were used in all further experiments.

A key feature of the allocator is to be able to direct transcriptional resources to different actuators. This requires multiple σ fragments that can bind to the core fragment to change its promoter affinity. These σ fragments need to be orthogonal, that is, they cannot cross-react with each other's promoters. Initially, we attempted to base the orthogonal σ fragments on a set of specificity loop mutations previously shown to generate orthogonal variants of full-length T7 RNAP (Temme *et al*, 2012a). These specificity loops are based on polymerases from the T3, K1F, and N4 phages. We tested the corresponding σ fragments and mutated promoters. Unfortunately, of these variants, only the σ fragment containing the T3 specificity loop and corresponding promoter (Fig 3C) generated an activity comparable to that of the T7 σ fragment (Fig 3D).

The σ fragments based on the K1F and N4 specificity loops did have some residual activity. This was used as a basis to apply errorprone PCR to the σ fragments to search for mutations that increase activity (Materials and Methods). One mutation was found for the K1F loop (K1FR: M750R) that recovered activity to a sufficient level, but similar efforts with the N4 loop proved unsuccessful (Supplementary Information Section III.A.). An additional σ fragment was built based on an orthogonal T7 RNAP variant (CGG-R12-KIR) that was identified from directed evolution experiments (Ellefson *et al*,

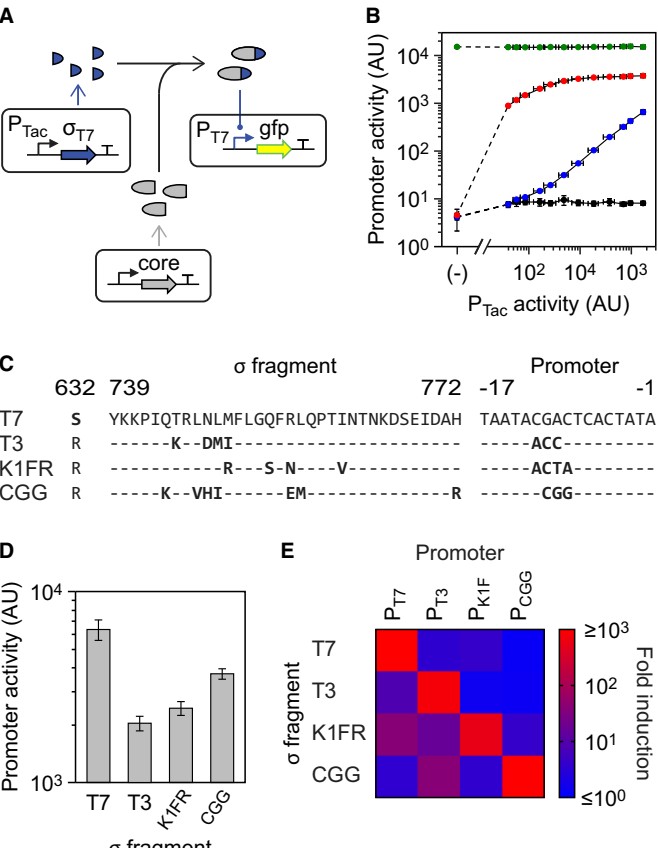

**Figure 3.** **Activation of the core fragment via σ fragments.**

A  A schematic of the induction system is shown; the core fragment is expressed at a constant level from a constitutive promoter.

B  The T7 σ fragment (SZ-601:883) is induced in the presence of different core fragments, and the activity of $P_{T7}$ is measured. Red and blue points show the induction in the presence and absence of the SynZIP, respectively (core fragments 1:601-SZ and 1:601). The activity of full-length T7* RNAP is shown as a positive control (green). A negative control with no core fragment is shown (black). The leftmost point (marked '(−)') represents cells that did not encode the T7 σ fragment. From left to right, the remaining points represent induction levels of: 0, 1, 2, 4, 6.3, 10, 16, 25, 40, 63, 100, and 1,000 μM IPTG.

C  The variations between the σ fragments and promoters are shown. Position 632 indicates the mutation made in T7* RNAP that reduces toxicity, and positions 739–772 show the DNA-binding loop.

D  The activities of each of the four σ fragments are shown with their cognate promoters when expressed to saturation (100 μM IPTG) with the core fragment.

E  The cross-reactivity of each σ fragment with each promoter is shown (100 μM IPTG induction of the σ fragments and constant core fragment expression). The underlying activity levels and variation for this assay are shown in Supplementary Fig S5.

Data information: For all graphs, the mean is shown for three independent assays performed on different days, with error bars showing standard deviation.

Source data are available online for this figure.

2013). This produced a comparable activity to the other σ fragments (Fig 3D). In total, four σ fragment variants (T7, T3, K1FR, and CGG) and cognate promoters were built. It is noteworthy that the σ fragments only differ in sequence by 5–10 amino acids (Fig 3C). Expression of each σ fragment with its cognate promoter and the

same level of core fragment shows that their activities fall into a similar range with less than a fourfold difference between the strongest (T7) and weakest (T3) σ fragments (Fig 3D). The four σ fragments were also found to be orthogonal (Fig 3E), and their expression to saturation with the core fragment does not lead to growth defects (Supplementary Fig S10).

## Setting and sharing the transcriptional budget

The expression level of the core fragment from the resource allocator sets the maximum number of active RNAPs in the synthetic system. This budget has to be shared between σ fragments that are expressed simultaneously (Fig 1C). To test this, we built a plasmid where the K1FR σ fragment is expressed from $P_{Tet}$ and the T3 σ fragment is expressed from $P_{Tac}$ (Fig 4A). By inducing the system with IPTG, the level of expression of the T3 σ fragment is varied while the K1FR σ fragment is maintained at a constant level ($P_{Tet}$ is uninduced but has leaky expression). In essence, this captures the scenario where one output of a controller is constantly on at a saturating level and then another output turns on and competes for the RNAP resource. To report how much of each type of polymerase complex is present in the system, reporter plasmids that express GFP from $P_{T3}$ and $P_{KIF}$ were used. The activity of the $\sigma_{T3}$:$P_{T3}$ and $\sigma_{K1FR}$:$P_{K1F}$ pairs are very similar (Fig 3D), making it possible to compare their expression levels.

Core fragment expression was driven by the $P_{J23105}$ promoter with RBSs of different strengths. Initially, a strong RBS was chosen that sets a high expression level of the core fragment (Fig 4B). The K1FR σ fragment utilizes the majority of the core fragment budget before the T3 σ fragment is induced. As the T3 σ fragment is induced, it competes for the core fragment. At high concentrations, it saturates the pool of core fragment, almost completely titrating it from binding to the K1FR σ fragment. The sum of the $P_{K1F}$ and $P_{T3}$ promoter activities (gray points) remains constant and is independent of the expression of either σ fragment. The competition experiment was repeated with the core fragment expressed at a lower level from a weaker RBS (Fig 4C). Importantly, the expression level of the K1F σ fragment and the induction of the T3 σ fragment remain unchanged. As before, the sum of activities from the $P_{T3}$ and $P_{K1F}$ promoters remains constant. Both of these competition systems are tolerated by cells with little growth impact at the induction levels used (Supplementary Fig S11).

The shapes of the curves are essentially identical when compared for high and low concentrations of the core fragment. The similarity is shown by plotting the $P_{T3}$ and $P_{K1F}$ promoter activities with low core fragment expression against their activities with high core fragment expression (Fig 4D). This results in a linear relationship, meaning that all promoter activities scale equally with the amount of core fragment expressed. The slope of this line indicates that the low level of core fragment yields approximately 36% of the activity compared to the high level. Hence, the budget is shared identically between the σ fragments at each core fragment expression level. This property means that the proportional outputs of the resource allocator can be set independently from the level of resource being produced.

To correct for the slight activity difference between the T3 and K1FR systems, we normalized the $P_{T3}$ and $P_{K1F}$ activity values by the activity when each individual σ fragment is expressed to saturation with the appropriate resource allocator (Fig 4E). Assuming

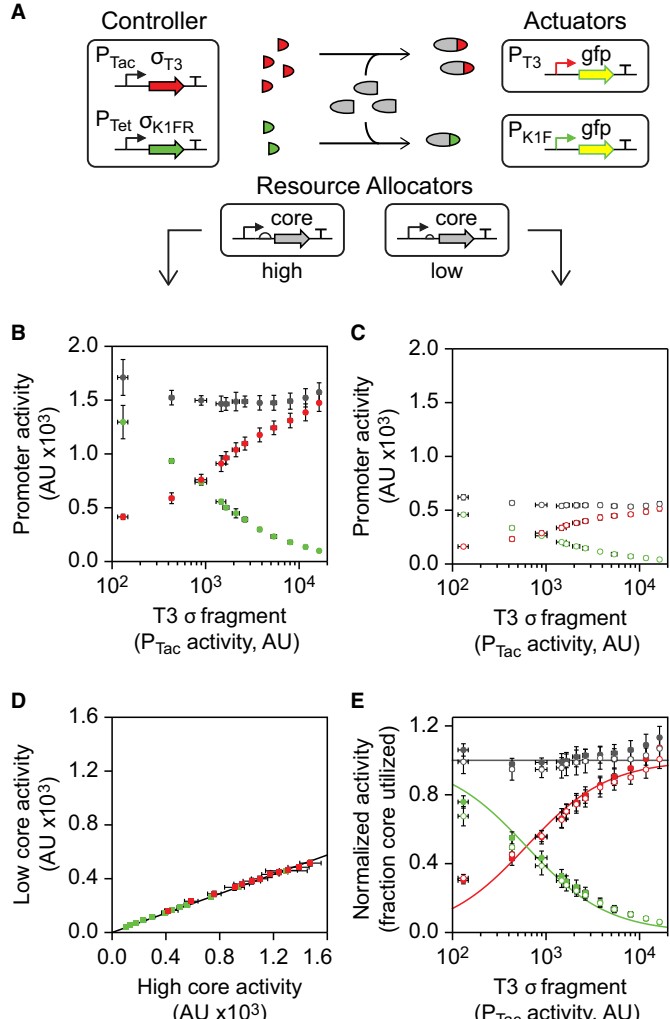

**Figure 4.  Competition between σ fragments to bind the core fragment.**

A   The genetic system used for the competition assays is shown. Two resource allocator plasmids were built that generate high and low core fragment expression levels via a strong or weak RBS and constitutive promoter.

B   Data for the high resource allocator are shown. The K1FR σ fragment was expressed at a constant level (no induction of $P_{Tet}$), and the T3 σ fragment was induced with 0, 2, 4, 6.3, 7.4, 8.6, 10, 13, 16, 20, 25, and 32 μM IPTG. The activities of $P_{T3}$ (red circles) and $P_{K1F}$ (green circles) were measured, and the sum of their activities computed (gray circles).

C   Data for the low resource allocator are shown, as in (B).

D   Each point represents promoter activity (red: $P_{T3}$, green: $P_{K1F}$) at a specific level of inducer. The x and y values show the activity with high and low levels of core fragment expression, respectively. The line shows a linear regression, with the intercept fixed to 0.

E   Each σ fragment was expressed to saturation (100 μM IPTG) with the high and low resource allocators, and the measured promoter activities were used to normalize the data shown in (B) and (C) (solid and hollow circles, respectively). The 'fraction core utilized' represents the proportion of the core fragment present in the system that is bound by either σ fragment, assuming a linear correlation with promoter activity.
The solid lines show a simplified model of competition fit to the normalized data.

Data information: For all graphs, the mean is shown for three independent assays performed on different days, with error bars showing standard deviation.
Source data are available online for this figure.

that promoter activity is linearly proportional to the number of active polymerases, these normalized values represent the proportion of the available core fragment bound by each of the σ fragments. A mathematical model of the system was built and its dynamics analyzed (Supplementary Information Section IV.B.). When the core fragment is fully saturated by σ fragments, the model predicts that the proportion of the core fragment bound by each σ fragment should depend solely on the relative expression levels of each σ fragment. The simplified model has only one parameter not measured in the normalized data set: the relative expression of the K1FR σ fragment (Supplementary Information Section IV.C, Equations 29-30). Fitting this parameter yields a good agreement between the theory and experimental data (Fig 4E, Supplementary Equations 31-33).

### Positive and negative regulation of the core fragment

The resource allocators shown in Figs 3 and 4 maintain a constant level of core fragment. It is desirable to be able to dynamically shift the budget up or down, for example, to control the maximum transcriptional capacity as a function of media or growth phase. To do this, we used additional splits and mutations to create positive and negative regulators. These regulators could also be used to design feedback or feedforward circuits to implement control algorithms that act on the signal from the controller plasmid to the actuators.

The negative regulator is based on a 'null' σ fragment that binds to the core fragment but does not support transcription. This functions to sequester the core fragment in the same way as an active σ fragment, making less of it available to the other competing σ fragments. Sequestration has emerged as a generalizable method to tune the threshold and ultrasensitivity of genetic circuits by setting a concentration of sequestering molecule that must be outcompeted before the circuit turns on (Buchler & Louis, 2008; Buchler & Cross, 2009; Chen & Arkin, 2012; Rhodius *et al*, 2013). The null fragment was identified by testing amino acid substitutions and deletions identified from the literature to disrupt T7 RNAP function (Bonner *et al*, 1992; Mookhtiar *et al*, 1991). These mutations were selected to disrupt transcription activity without impacting the ability of the σ fragment to bind and sequester the core fragment (Supplementary Table S4). Based on the screen, we identified the Y638A mutation in the CGG σ fragment as having the strongest effect when sequestering the core fragment. This fragment was confirmed to carry no residual activity for its original promoter (Supplementary Fig S6).

A system was constructed to test the ability of the null fragment to titrate the core fragment and reduce its availability to the σ fragments (Fig 5A). For this, the σ fragments were expressed using a constitutive promoter derived from $P_{J23119}$ and the null fragment was placed under $P_{Tac}$ IPTG-inducible control on a separate plasmid. When expressed with the T7 σ fragment, the null fragment decreases the activity from $P_{T7}$ as it is induced (Fig 5B). The null fragment is able to compete with all of the σ fragments and reduces each of their activities by at least tenfold when fully induced (Fig 5C).

The positive regulator is based on further splitting the core fragment at the most N-terminal split site (Fig 2B and D). This divides the core fragment into two pieces: a short 67 amino acid 'α fragment' and a larger 586 amino acid 'β core fragment' (including the SynZIP). The α fragment can be expressed separately and is required

for activity. It can be used to modulate the fraction of the polymerase pool that is active. Note that it still does not enable more transcriptional activity than is set by the amount of β core fragment that is expressed. Thus, the maximum can be set and then the α fragment used to modulate the amount that is available at any given time.

A system was constructed to assay the α fragment's ability to regulate the polymerase budget (Fig 5D). The β core fragment is expressed from the $P_{J23105}$ constitutive promoter on a low copy plasmid, while the T7 σ fragment is expressed from a constitutive promoter derived from $P_{J23119}$ on a high copy plasmid. The α fragment is expressed from $P_{Tac}$. There is no T7 RNAP activity without the α fragment and activity increases as it is induced (Fig 5E).

### Coupling RNAP activity to the concentration of arbitrary α fragment tagged proteins

Since the α fragment is relatively small (67 aa) and required for polymerase function, we hypothesized that it would be useful as a protein tag to activate transcription proportional to the level of an arbitrary protein of interest. While the C-terminus of T7 RNAP catalyzes transcription and is highly sensitive to alteration, the N-terminus (where the α fragment is located) is much more tolerant to modifications (Dunn *et al*, 1988). The α fragment was fused to proteins of interest via a GGSGG flexible linker. Fusion to either the N- and C-terminus of RFP or GFP makes polymerase activity responsive to the level of fluorescent protein expression (Fig 5F and Supplementary Fig S7). This may be used to tag proteins in a synthetic system or the host, enabling the readout of an internal or cell state.

### Application of the α fragment to compensate for differences in copy number

A challenge in building genetic systems is that regulatory parts will change their activity depending on the copy number of the system. For example, a constitutive promoter will produce a high level of expression when it is placed on a high copy plasmid and a low level of activity with placed at single copy on a bacterial artificial chromosome (Kittleson *et al*, 2011). The α fragment could be used to regulate the activity of the polymerase to adjust the activity of promoters and compensate for the copy number at which they are carried due to different plasmid origins (or in the genome). The idea is to combine the phage promoter(s) with an expression cassette including the α fragment that is expressed at a level inversely proportional to the copy number (Fig 5G). In other words, a strong promoter and RBS would be selected to drive the expression of the α fragment from a low copy plasmid and vice versa.

Plasmids were constructed on pSC101 and pUC backbones that contain a $P_{T7}$ promoter driving GFP expression and a α fragment expression cassette. We mutagenized the RBSs and altered the promoters and start codon of the α fragment expression cassettes to identify a strong cassette that would be carried on the pSC101 plasmid and weak cassette that would be carried on the pUC plasmid (Materials and Methods). With these different levels of α fragment expression, we were able to achieve nearly identical activities for $P_{T7}$ in the different plasmid contexts when they are used with the β core fragment (Fig 5H). In contrast, when the plasmids are used

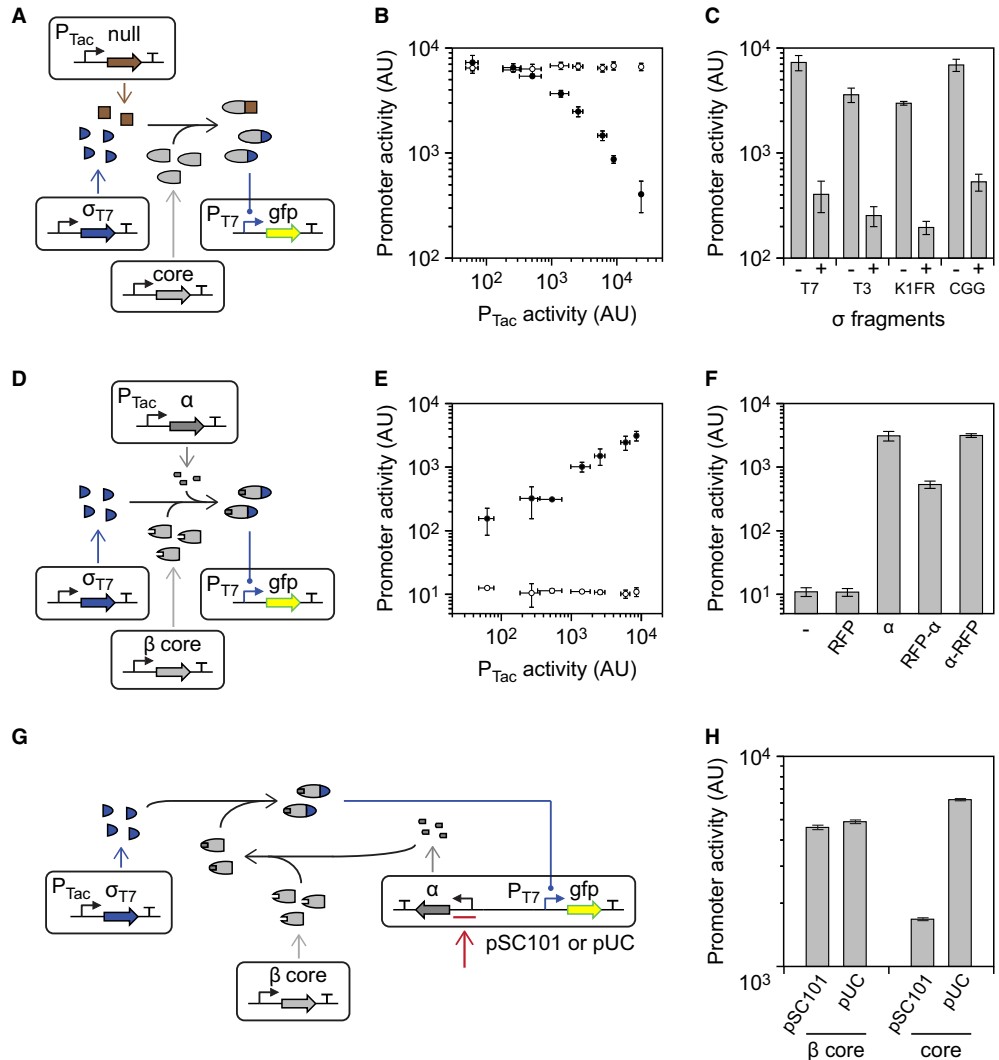

**Figure 5. Positive and negative post-transcriptional regulation of the core fragment.**

A   Null fragment sequestration of the core fragment.

B   The core fragment and T7 σ fragment are expressed constitutively, while null fragment expression is induced from $P_{Tac}$ (induction from left to right is: 0, 2, 4, 10, 16, 25, 40, and 1000 µM IPTG). The effect of the expression of the null fragment on $P_{T7}$ activity is shown as black circles. The activity of $P_{T7}$ under the same conditions lacking the inducible null fragment cassette is shown as white circles.

C   The null fragment is shown in competition with each of the four σ fragments. Data are shown when the null fragment is uninduced (−, 0 µM IPTG) and induced (+, 1000 µM IPTG).

D   Activation of the β core fragment through the expression of the α fragment.

E   The impact of expressing the α fragment from the $P_{Tac}$ promoter is shown. The black and white circles show induction in the presence and absence of the α fragment cassette, respectively (from left to right: 0, 2, 4, 10, 16, 25, and 40 µM IPTG). The high level for uninduced is due to leaky expression from $P_{Tac}$.

F   The ability of α fragment : RFP fusions to complement the β core fragment (with the T7 σ fragment) is shown. From left to right: (−), no inducible cassette; RFP, expression of unmodified RFP; α, expression of free α fragment; RFP-α, expression of a C-terminal fusion of α fragment to RFP; α-RFP, expression of an N-terminal fusion. Each system was induced with 40 µM IPTG.

G   A genetic system is shown that uses α fragment expression from a constitutive promoter to compensate for the effects of differences in copy number. A strong constitutive promoter and RBS controlling α expression (red arrow) are selected at low copy (pSC101), while a weaker promoter and RBS are used at high copy (pUC).

H   Data are shown for a pair of pSC101 and pUC plasmids carrying tuned α fragment cassettes and a $P_{T7}$ promoter driving GFP. 'β core' indicates that the β core fragment and T7 σ fragment are co-expressed. 'core' indicates that the core fragment and T7 σ fragment are co-expressed.

Data information: For all graphs, the mean is shown for three independent assays performed on different days, with error bars showing standard deviation.

with the full core fragment, which does not need the α fragment to function, high expression is seen from the high copy pUC backbone and low expression is seen from the low copy pSC101 backbone.

One of the values of this approach is that it enables actuators that require multiple phage promoters to be moved to different copy number contexts without having to change and rebalance each of the promoters. For example, actuators that produce deoxychromo-viridans, nitrogenase, and lycopene require 2, 4, and 5 phage promoters (Temme *et al*, 2012a,b). These could be moved to different copy number backbones without changing their genetics by

changing the expression level of the α fragment from that backbone. One can also imagine harnessing feedback or feedforward loops that self-adjust the level of α fragment to maintain constant promoter activity independent of context, similar to systems that have been implemented in mammalian cells (Bleris *et al*, 2011).

## Discussion

As a means to organize and control large genetic engineering projects, we propose to introduce a separate resource allocator module. The allocator is responsible for providing resources that are orthogonal to those required by the host for growth and maintenance. To that end, this manuscript focuses on budgeting transcriptional resources through the control of phage polymerase activity and promoter specificity. Thinking ahead, this approach can be extended to budget additional resources. For example, translational resources could be incorporated by controlling a orthogonal rRNA (Rackham & Chin, 2005; An & Chin, 2009) (specific to RBSs only in the synthetic system) or even introducing an entire second ribosome. Extending this idea, it may be possible to incorporate orthogonal tRNAs (Liu *et al*, 1997; Chin, 2014), DNA replication machinery (Ravikumar *et al*, 2014), protein degradation machinery (Grilly *et al*, 2007), carbon precursors (Pfeifer *et al*, 2001), and organelle structures (Moon *et al*, 2010; Bonacci *et al*, 2012). While this never completely decouples the synthetic system from the host, it systematically reduces its dependence on host resources and genetic idiosyncrasies. This approaches the concept of a 'virtual machine' for cells, where synthetic systems would bring all of the necessary cellular machinery with them. This concept will become critical as designs become larger, moving toward the scale of genomes and requiring the simultaneous control over many multi-gene actuators.

This work demonstrates an incredible tolerance of the T7 RNAP structure for division into multiple proteins without disrupting its function. To our knowledge, this is the first time that a protein has been artificially divided into four fragments that can be functionally co-expressed. This tolerance is surprising because T7 RNAP is known to undergo large-scale conformational changes as it proceeds from promoter binding to transcription elongation (Ma *et al*, 2002; Guo *et al*, 2005). The residues involved in these conformational changes occur toward the N-terminal region but are distributed across the first three fragments of the 4-fragment polymerase (Fig 2E). All of the RNAP split points were discovered simultaneously using a new experimental method, which we refer to as a 'splitposon'. This approach is faster, simpler, and produces more accurate split proteins than previous methods. Split proteins have applications in genetic circuits (Shis & Bennett, 2013; Mahdavi *et al*, 2013), plasmid maintenance with fewer antibiotics (Schmidt *et al*, 2012), and biosensors (Johnsson & Varshavsky, 1994; Galarneau *et al*, 2002; Hu & Kerppola, 2003; Michnick *et al*, 2007; Camacho-Soto *et al*, 2014).

The fragments of T7 RNAP are used to implement regulatory control. A C-terminal fragment contains the DNA-binding loop and we demonstrate that fragments with different specificities can direct the RNAP to different promoters. For this reason, and because of its size, we draw a loose analogy to the role of σ factors in native prokaryotic transcription. However, there are notable differences between our σ fragments compared to natural σ factors. First, core *E. coli* RNAP binds to DNA in a non-specific manner and this is titrated away by the σ factors (Grigorova *et al*, 2006; Bratton *et al*, 2011). It is unlikely that our T7 RNAP core fragment binds to DNA. Second, a prokaryotic σ factor only recruits the RNAP to the promoter and once transcription initiation is complete, the σ factor dissociates during transcription (Travers & Burgess, 1969; Raffaelle *et al*, 2005). Thus, the ratio of σ factors to core RNAP is low (~50%) because they only have to compete to bind to free (non-transcribing) polymerase (Ishihama, 2000). Our system requires larger ratios, because the σ fragments must remain associated with the core fragment during transcription. Third, while the size of a σ factor and the σ fragment are about the same, their 3-dimensional structure and mechanism of binding to core and DNA are different (Vassylyev *et al*, 2002). Finally, recent results suggest that the *B. subtilis* core RNAP is shared by σ factors in time as opposed to concentration (Levine *et al*, 2013). In other words, the σ factors pulse in a mutually exclusive manner to take turns fully utilizing the pool of core RNAP. In contrast, our σ fragments compete for the core fragment following mass action kinetics. This is similar to the previous understanding, where differences in σ factor binding affinities are a means that cells prioritize and order different responses (Lord *et al*, 1999; Maeda *et al*, 2000; Grigorova *et al*, 2006).

Resource allocation also occurs in natural regulatory networks. In bacteria, alternative σ factors can redirect RNAP to different condition-specific promoters. Factors such as ppGpp and 6S RNA also regulate the pool of active free RNAP (Jensen & Pedersen, 1990; Wassarman & Storz, 2000; Klumpp & Hwa, 2008). Using up this resource has been observed and shown to result in a slower growth rate (Farewell *et al*, 1998). Further, the competition between σ factors for core RNAP has been quantified (De Vos *et al*, 2011; Grigorova *et al*, 2006). Keren and co-workers measured the activity of thousands of native *E. coli* and *S. cerevisiae* promoters under different environmental conditions (Keren *et al*, 2013). They found that while changes in conditions have a global impact on many promoters, they shift by a linear factor that is characteristic of each condition. This factor ranges from 0.51 to 1.68 with M9 + glucose being the reference condition. They found that a simple model that treats overall promoter activity as a fixed resource explains their data. Overall promoter activity is equivalent to the total active RNAP concentration that forms the backbone of our resource allocator and the ratio of 0.36 shown in Fig 4D is analogous to their linear factor when moving from the high to the low resource allocator.

In the context of synthetic signaling networks, retroactivity occurs when downstream regulation impacts an upstream process. For example, the titration of ribosomes or proteases by one branch of the network can influence the network as a whole (Cookson *et al*, 2011). This is viewed as an undesirable effect that must be buffered against in order to maintain computational integrity (Del Vecchio & Murray, 2014). In contrast, the resource allocator harnesses retroactivity in order to budget transcription to different pathways without surpassing a limit. As an allocation mechanism, retroactivity is an ideal means of distributing a budgeted resource. Currently, this is limited to dividing the core fragment among the σ fragments in a way that is proportional to their expression levels. Building on this, more complex dynamics could be introduced that implement signal processing between the output of the controller plasmid and the actuators that are being regulated. For instance, it may be desirable to control several actuators via a mutually exclusive or analog relationship, for example to slow down a metabolic pathway as a

molecular machine is being built. Other actuators may require graded or ultrasensitive responses, for example the all-or-none commitment to flagellum construction versus simply changing the level of an enzyme. The toolbox presented in this paper provides a means to rationally design such control that can be implemented on the signal from the output of circuitry encoded on a controller to the actuators.

# Materials and Methods

## Strains and media

*Escherichia coli* DH10B was used for all routine cloning and characterization. ElectroMAX competent cells (Life Technologies) were used for library cloning steps as noted. LB-Miller media was used for assays and strain propagation, 2YT media was used for strain propagation, and SOC media was used for transformation recovery. Antibiotics were used as necessary for plasmid maintenance, with ampicillin at 100 μg/ml, spectinomycin at 100 μg/ml, kanamycin at 50 μg/ml, and chloramphenicol at 17 μg/ml. IPTG (isopropyl β-D-1-thiogalactopyranoside) was used as an inducer at concentrations up to 1 mM.

## Plasmids and parts

Plasmids with the ColE1 origin were based off of the plasmid pSB1C3 from the Registry of Standard Biological Parts, which has a pUC19 (Yanisch-Perron *et al*, 1985) derived origin. Plasmids with the pUC origin were based off of a pUC19 (Yanisch-Perron *et al*, 1985) vector. Plasmids with the p15A* origin were based off of plasmid pSB3C5 (Shetty *et al*, 2008) from the Registry. This origin appears to maintain at a higher copy number than standard for p15A. Plasmids with the pSC101 origin were based on pUA66 (Zaslaver *et al*, 2006). Plasmids with the BAC origin were based on pBACr-Mgr940 (Anderson *et al*, 2007) (BBa_J61039), which has an F plasmid derived origin. A $P_{Tac}$ promoter system derived from pEXT20 (Dykxhoorn *et al*, 1996) modified to contain a symmetric LacI binding site or a shortened version of this expression system was used in all systems that required inducible expression. Constitutive protein expression was driven by promoter $P_{J23105}$ (BBa_J23105) or $P_{J23109}$ (BBa_J23109), by a modified $P_{Tet}$ expression system (Moon *et al*, 2012) (uninduced), and by promoters selected from libraries derived from $P_{J23119}$ (BBa_J23119) through degenerate PCR. RBSs were either generated using the RBS calculator, taken from the Registry (BBa_B0032 and BBa_B0034 (Elowitz & Leibler, 2000)), or selected from libraries generated using degenerate PCR. The RiboJ insulator (Lou *et al*, 2012) was used between $P_{Tac}$ or $P_{Tet}$ and the RBS in all constructs when titrations curves were run. mRFP1 (Campbell *et al*, 2002) and sfGFP (Pédelacq *et al*, 2006) were used as fluorescent reporters. Representative plasmid maps are shown in Supplementary Figs S2, S9, and S13 through S19. A list of new plasmids is given in Supplementary Table S6. Select constructs from this study will be made available online through Addgene (http://www.addgene.org/Christopher_Voigt/).

## Bisection mapping T7 RNA polymerase

The splitposon was generated by modifying the HyperMu <KAN-1> transposon (Epicentre Biotechnologies). Examining previously described variants of the MuA transposon system (Goldhaber-Gordon *et al*, 2002; Poussu *et al*, 2004, 2005; Jones, 2006; Hoeller *et al*, 2008), a number of terminal bases were identified that could be altered while maintaining transposition activity. The RBS calculator (Salis, 2011) was used to design a strong terminal RBS and start codon while staying within these alterations. This modified end was combined with a previously built end containing terminal stop codons (Poussu *et al*, 2005). A $P_{Tac}$ promoter and constitutive LacI expression cassette were inserted into the transposon to drive transcription at the end with the RBS and start codon. Finally, point mutations were made to remove restriction sites that would interfere with downstream cloning steps. A region of the T7* RNA polymerase CDS encoding aa 41–876 was flanked by BsaI sites in a ColE1 AmpR backbone. The splitposon (KanR) was transposed into this plasmid with MuA transposase (300 ng target DNA, 200 ng transposon, MuA buffer, 1.1 U HyperMuA transposase (Epicentre Biotechnologies), 30°C 8 h, 75°C 10 min), DNA clean and concentrated (Zymo), electroporated into ElectroMAX cells and plated on LB + Kan/Amp plates to obtain > 700,000 colonies. The colonies were scraped from the plates, pooled, and miniprepped to obtain DNA of the transposon insertion library. The transposon insertion library was digested with BsaI, run on an agarose gel, and a band of ~5.7 kb (representing the section of the T7 CDS plus transposon) was excised, gel-purified (Zymo), and DNA clean and concentrated. A plasmid containing an inducible $P_{Tac}$ system and the remainder of the T7 CDS (aa 1–40 and 877–883) with internal BsaI sites on a p15A* SpecR backbone was digested with BsaI and the size-selected fragment ligated into it. This reaction was DNA clean and concentrated, electroporated into ElectroMAX cells plated on LB + Spec/Kan plates to obtain > 600,000 colonies, and the colonies were scraped, pooled, and miniprepped as before to obtain the bisected library. This library was electroporated into *E. coli* DH10B cells with a plasmid containing a $P_{T7}$-RFP cassette on a pSC101 CamR backbone (Nif_489 (Temme *et al*, 2012a)), plated on LB + Spec/Kan/Cam, and visually red colonies were picked after 16 h of growth for analysis in liquid media. More information on the splitposon method and T7 RNAP bisection mapping are included in Supplementary Information Sections I and II.

## Assay protocol

All promoter activity assays except the initial assay of T7 bisection mapping were performed as follows. Cells containing the plasmids of interest were inoculated from glycerol stocks into 0.5 ml LB-Miller media plus antibiotics in a 2-ml 96-deepwell plate (USA Scientific) sealed with an AeraSeal film (Excel Scientific) and grown at 37°C, 900 rpm overnight (~14–16 h) in a deepwell shaker. These overnights were diluted 200-fold into 150 μl LB-M with antibiotics plus varying concentrations of IPTG in 300-μl 96-well V-bottom plates (Thermo Scientific Nunc) sealed with an AeraSeal film and grown at 37°C, 1,000 rpm for 6 h. 5 μl of each sample was removed and diluted in 195 μl PBS + 2 mg/ml kanamycin to halt protein production. Cells diluted in PBS were either characterized immediately with flow cytometry or stored at 4°C until characterization. The initial T7 bisection mapping assays were performed similarly except the overnight cultures were grown in 2YT, and the overnight cultures were diluted 1:10 into 150 μl induction media.

## Flow cytometry characterization

All fluorescence characterization was performed on a BD LSR Fortessa flow cytometer with HTS attachment and analyzed using Flow-Jo vX (TreeStar). Cells diluted in PBS + kanamycin were run at a rate of 0.5 µl/s until up to 100,000 events were captured (at least 50,000 events were recorded in all cases). The events were gated by forward scatter and side scatter to reduce false events and by time to reduce carry-over events. Gating was determined by eye and was kept constant for all analysis within each triplicate experiment. For all assays except the initial characterization of T7 bisection mapping, the geometric mean value of fluorescence was calculated for each sample, using a biexponential transform with a width basis of −10.0 to allow calculations with negative values. Finally, white-cell fluorescence measured concurrently from cells lacking fluorescent protein was subtracted from measured fluorescence to yield the Promoter activity (AU) values presented in the figures. The initial T7 bisection mapping assay was characterized identically, except that white-cell values were not subtracted.

Where fold induction calculations were required, fluorescence measurements were made of cells containing the appropriate reporter construct and lacking a functional polymerase, grown in the same conditions as the test cells. The fold induction is reported as the ratio of the white-cell-corrected test cell fluorescence to the white-cell-corrected fluorescence of the reporter-only cells.

To obtain relative expression levels for the polymerase fragments driven by $P_{Tac}$, constructs were made that express GFP after $P_{Tac}$ and RiboJ (Supplementary Fig S9). For each assay, cells with this construct were induced under the same conditions as the test cells, and their fluorescence measured (Supplementary Fig S8). The $P_{Tac}$ activity value in each plot represents the geometric mean white-cell-corrected fluorescence of these cells for that assay, and the horizontal error bars show the standard deviation of those measurements.

## Measuring the growth impact of split polymerase expression

Cells containing the plasmids of interest were inoculated from colonies on agar plates into 0.5 ml LB-Miller media plus antibiotics in a 2-ml 96-deepwell plate, sealed with an AeraSeal film, and grown at 37°C, 900 rpm overnight (~14–16 h) in a deepwell shaker. These overnights were diluted 200-fold into 150 µl LB-M with antibiotics plus varying concentrations of IPTG in 300-µl 96-well V-bottom plates, sealed with an AeraSeal film, and grown at 37°C, 1,000 rpm for 6 h. 20 µl of each sample were added to 80 µl LB in a 96-well optical plate (Thermo Scientific Nunc), and the $OD_{600}$ of each diluted sample was measured using a BioTek Synergy H1 plate reader. These measurements were normalized by dividing by the $OD_{600}$ of samples containing plasmids with the same backbones but expressing none of the proteins of interest (polymerase fragments or GFP) at each level of IPTG induction. Growth data are shown in Supplementary Figs S10, S11 and S12.

## Error-prone PCR of σ fragment variants

Sections of the K1F and N4 T7 RNAP variants (Temme *et al*, 2012a) were amplified using GoTaq (Promega) in 1× GoTaq buffer plus MgCl₂ to a final concentration of 6.5 mM $Mg^{2+}$. The amplified fragments were cloned into a σ fragment expression plasmid including any necessary flanking RNAP sequence and the N-terminal SynZIP 18 domain. These mutated σ fragments were expressed with the core fragment and the appropriate promoter driving GFP. Colonies with visually improved GFP production were picked from plates, re-assayed to confirm activity, and sequenced to identify their mutations (Supplementary Tables S2 and S3). Promising variants were reconstructed to isolate their effects and the resulting new σ fragments assayed for activity.

## Tuning α fragment expression to compensate for copy number

An α fragment expression cassette consisting of the constitutive promoter $P_{J23105}$, RiboJ, and B0032 RBS driving the α fragment was inserted in the reverse direction before the $P_{T7}$: GFP cassette on a pSC101 reporter plasmid. These two cassettes were also inserted into a pUC19 backbone, with the weaker constitutive promoter $P_{J23109}$ and start codon (GTG instead of ATG) in the α fragment cassette. Degenerate PCR was used to randomize the RBS in each plasmid at five nucleotides, and the resulting libraries were screened for fluorescence in the presence of the $\sigma_{T7}$ and either core or β core fragments. Sets of pSC101 and pUC plasmids were selected that had similar levels of activity with the β core fragment, but retained different levels of activity with the core fragment. These plasmids were isolated, sequenced, re-assayed, and the pair of pSC101 and pUC plasmids with the closest levels of expression in the presence of the β core fragment was selected.

**Supplementary information** for this article is available online: http://msb.embopress.org

## Acknowledgements

This work was supported by the United States Office of Naval Research (N00014-13-1-0074), the United States National Institutes of Health (5R01GM095765), and the US National Science Foundation Synthetic Biology Engineering Research Center (SA5284-11210). THSS was supported by the National Defense Science & Engineering Graduate Fellowship (NDSEG) Program and by a Fannie and John Hertz Foundation Fellowship.

## Author contributions

THSS and CAV conceived of the study. THSS carried out experiments. AJM and ADE developed the CCG T7 RNAP variant. THSS and EDS modeled and analyzed the system. THSS and CAV wrote the manuscript with input and contributions from all of the authors.

## Conflict of interest

A patent application has been filed on some aspects of this work, with THSS and CAV as inventors.

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
