## [Review Process File · Molecular Systems Biology]

A 'resource allocator' for transcription based on a highly fragmented T7 RNA polymerase

Thomas Segall-Shapiro, Adam J Meyer, Andrew D. Ellington, Eduardo D. Sontag and Christopher A. Voigt

Corresponding author: Christopher A. Voigt, Massachusetts Institute of Technology

Review timeline:	Submission date:	21 March 2014
	Editorial Decision:	25 April 2014
	Revision received:	05 June 2014
	Accepted:	24 June 2014

Editor: Maria Polychronidou

Transaction Report:

1st Editorial Decision

25 April 2014

Thank you again for submitting your work to Molecular Systems Biology. We have now heard back from two of the three referees who agreed to evaluate your manuscript. Since their recommendations are very similar, I prefer to make a decision now rather than further delaying the process. As you will see from the reports below, the reviewers acknowledge that the presented findings are potentially interesting for synthetic biology applications. However, they raise a series of concerns, which should be carefully addressed in a revision of the manuscript.

Without repeating all the points listed below, most of the reviewers' comments refer to the need to provide additional explanations and/or clarifications and to include minor modifications. Thank you for submitting this paper to Molecular Systems Biology.

Reviewer #1:

This manuscript represents a large body of work and is an exciting new development in synthetic biology. One challenge faced by synthetic biologists in constructing synthetic gene networks is the variation in gene introduced by the host. One strategy is to use an orthogonal transcription system such as T7 RNAP driven transcription. The use of these systems is precluded by the lack of regulation and diversity of open reading frames. To address this, Segal-Shapiro et al functionally split T7 RNA Polymerase (T7 RNAP) into multiple fragments and lay the groundwork for a 'virtual machine' transcription environment.

The authors functionally split T7 RNAP in two places to create a N-terminal 'activator', central

'core', and c-terminal 'sigma' fragment, to facilitate a synthetic transcriptional virtual environment. The authors first split T7 RNAP into a 'core' and a 'sigma' domain where activity and specificity are on separate protein fragments. In doing so, transcription will only occur if both fragments are present. They go on to demonstrate that the fusion of heterodimerization domains increases the transcriptional activity of the split protein. The authors then create orthogonal open reading frames expressed by their split protein by creating mutant sigma domains that bind to modified T7 promoters. They found modifications previously described to modify the specificity of full length T7 RNAP are not necessarily transposable to their version of split T7 RNAP. To address this, they evolved three additional versions of their sigma fragment, two of which orthogonally recognize mutant T7 promoters while the third acts as a repressor by sequestering 'core'. The authors next split T7 RNAP a third time in the core domain to create an 'activator' fragment. In this case, transcription only occurs if 'activator', 'core', and 'sigma' are each present. Controlling the expression of the N-terminal activator fragment can then activate transcription from a T7 promoter. The authors go on to characterize some properties and potential applications of transcriptional systems based on fragmented T7 RNAP. The transcriptional activity of split T7 RNAP is limited by the availability of core fragment. Utilizing singly split T7 RNAP, they demonstrate that by limiting the availability of 'core' fragment, transcription by T7 RNAP can be globally limited. They suggest that this can be an effective strategy for maintaining sub-toxic levels T7 RNAP driven transcription. Furthermore, by splitting the protein three ways, and thereby making available the alpha 'activator' fragment, two additional features become possible. In making the expression of alpha fragment proportional to the reporter output, the authors suggest a method for compensating for variations in plasmid copy number. In addition, the authors demonstrate the alpha fragment can be fused to other protein, making it possible to correlate in T7 RNAP transcriptional activity in vivo protein levels.

Overall, this is a very nice piece of work. However there are some points that should be addressed before I can recommend publication in MSB.

1. Whenever fragments of a protein are expressed, the question of overexpressing possibly misfolded protein, which may be toxic to the cell, is a concern. Is there a reduction in OD associated with the expression of multi-fragmented T7 RNAP? Furthermore, splitting a protein generally introduces instability into a protein, disrupting enzymatic activity. Is there a reduction in activity associated with T7 RNAP fragment in two to three places as described in this document?
2. The authors point out in their 'splitposon' the MuA transposon introduces a random amino acid onto the N-terminal of the C-terminal fragment of the newly split protein. Do the split fragments of multi-fragmented T7 RNAP contain the random AA added by the MuA transposon or are they taken out in the final version of the split protein?
3. While in figure 2C the authors demonstrate the activity of split T7 RNAP with SZ-18 on the C-terminal fragment alone has been demonstrated they do not show the activity when SZ-17 is attached to the N-terminal fragment alone. Does the split protein tolerate the lone heterodimerization domain on the N-terminal fragment as it does on the C-terminal fragment?
4. In figure 3E, the authors should reference the bar graph in the supplement that the heatmap is based on in the figure caption.
5. In most cases the authors show transcription by fragmented T7 RNAP driven to saturation by overexpression of a particular fragment of the polymerase. They do not do this in assaying the effect of alpha fragment in figure 5e. While T7 RNAP driven transcription approaches saturation, they do not show what happens when even more alpha fragment is expressed. Will transcription by fragmented T7 RNAP saturate when alpha fragment is overexpressed?

Reviewer #2:

This paper presents a synthetic biology study constructing a T7 RNA polymerase-based gene expression system that implements switchable genetic programs analogous to the different sigma-factor dependent gene switches for the host RNAP. T7 RNAP, a single-subunit polymerase is split into subunits that need to be co-expressed in order to function in gene expression. To that end a new method, termed splitposon is introduced. Variants of the DNA-binding subunits (sigma-like fragment) are constructed to obtain orthogonal (but competing) expression systems. The authors call

this system a resource allocator, as it functions by distributing a common resource (the polymerase core) and thus the total transcription activity among several genes or sets of genes.

This is an excellent study that addresses a highly timely issue, the allocation of resources and the coupling of gene expression to the cellular background and the availability of molecular machinery using an original synthetic approach. It is easy to imagine systems with more complex behaviors to be constructed based on this method. In addition, the very fact that the polymerase is still functional when split into 4 pieces is quite remarkable in itself.

Thus, overall I am very much in favor of publishing this paper after minor revisions. Below are a few comments that could be addressed to improve the paper.

- 1) In Fig. 2c-e, I would suggest to show the case of the original full T7 (T7*) RNAP for comparison of the transcriptional activity. This is done in Fig. 3b, but it would be useful here as well.
- 2) Does any of the systems constructed here affect the host cell, e.g its growth rate? Does this impose a limitation to the expression of for example the core fragment?
- 3) The core fragment appears to be always limiting in the cases studied here. While this is certainly desirable from the application point of view, it would be a nice control supporting the overall picture to show a case, where the core fragment is not limiting, for example in the competition assay (Fig. 4) or for the repressor systems (sigma_null, fig. 5a,c).
- 4) In the latter case, this control would also provide clearer evidence that repression indeed functions by sequestering the core part and not by some other mechanism. At least a good argument in support of the mechanism should be given.
- 5) I am not sure that the term 'resource allocator' as used in fig. 1 (for the part of the system encoding the core fragment) is appropriate. This part only sets the overall level of the limiting resource, but allocation to different actuators, the eventual task, is a collective property of the whole system including the sigma-fragment encoding parts. I would thus rather call the whole system a resource allocator and refer to the core-encoding subsystem as the limiting resource.
- 6) Overall, I think Fig. 1 could be improved. It is a bit confusing that fig. 1b and c are actually only very briefly discussed much later in the paper, maybe this could be moved to the discussion section (with a separate figure) or to the supplement.

1st Revision - authors' response

05 June 2014

The manuscript has been edited and new experiments have been added to address the reviewer suggestions. Major changes include:

The description of the mathematical model has been expanded significantly, including a derivation of the steady-state behavior and dynamic analysis presented in the Supplementary Information. This impacts the conclusions drawn from the model and allows a fit to the data in Figure 4E.

Growth experiments have been performed showing the impact of the expression of T7 RNAP fragments on cell health. This has been included in Supplementary Information Section III.F.

We have added data to show that the split T7 RNAP tolerates a SynZIP coil added to the N-terminal fragment alone and have re-generated the data in Figure 2C with a focus on lower induction levels to better highlight the increase in split polymerase activity from the addition of both SynZIP coils.

Source data files have been submitted for the key datasets that underlie the main figures, including Figures 2B, 3E, and 4.

In addition, we have been able to modify the text or add experiments to address all of the reviewer comments.

Reviewer #1:

1. *Whenever fragments of a protein are expressed, the question of overexpressing possibly misfolded protein, which may be toxic to the cell, is a concern. Is there a reduction in OD associated with the expression of multi-fragmented T7 RNAP?*

We have added a section to the Supplementary Information (Supplementary Information Section III.F.) where we measure the growth impact of expressing fragmented and multi-fragmented T7 RNAP.

Splitting a protein generally introduces instability into a protein, disrupting enzymatic activity. Is there a reduction in activity associated with T7 RNAP fragment in two to three places as described in this document?

It is difficult to directly answer this question because changes in activity can be due to expression, stability, toxicity, or functional changes due to breaking the backbone. In our experience, the 2- and 3- split T7 RNAP variants are clearly less active than wild-type, but because the latter manifests in toxicity, they can effectively yield equivalent or even higher maximum levels of expression when expressed from similar constructs. Comparatively, we see a ~5-fold decline in maximum activity moving from the 3-piece T7 RNAP to the 4-piece version at high induction, but the causes and true extent of this activity loss are unclear (Figure 2D-E).

2. *The authors point out in their 'splitposon' the MuA transposon introduces a random amino acid onto the N-terminal of the C-terminal fragment of the newly split protein. Do the split fragments of multi-fragmented T7 RNAP contain the random AA added by the MuA transposon or are they taken out in the final version of the split protein?*

Yes, all fragments used later in the paper match those discovered by the initial 'splitposon' method. We have clarified this point in the Results section.

3. *While in Figure 2C the authors demonstrate the activity of split T7 RNAP with SZ-18 on the C-terminal fragment alone has been demonstrated they do not show the activity when SZ-17 is attached to the N-terminal fragment alone. Does the split protein tolerate the lone heterodimerization domain on the N-terminal fragment as it does on the C-terminal fragment?*

We have re-run the assay in Figure 2C including a version with SZ17 on the N-terminal fragment alone, and it is tolerated by the system. While performing these new assays, we also lowered the induction level of the polymerase fragments (from 10 μ M to 4 μ M IPTG) in order to better show the increased activity of the split T7 RNAP with both coils.

4. *In Figure 3E, the authors should reference the bar graph in the supplement that the heatmap is based on in the figure caption.*

We have made this change.

5. *In most cases the authors show transcription by fragmented T7 RNAP driven to saturation by overexpression of a particular fragment of the polymerase. They do not do this in assaying the effect of alpha fragment in Figure 5e. While T7 RNAP driven transcription approaches saturation, they do not show what happens when even more*

alpha fragment is expressed. Will transcription by fragmented T7 RNAP saturate when alpha fragment is overexpressed?

We do not believe that the overexpression of the α fragment will effectively saturate the β core fragment. In fact, overexpression of the σ fragment also does not saturate the core fragment unless the SynZIP domains are included to enhance the protein-protein interaction (as can be seen in Figure 3B). There is not another antiparallel, orthogonal pair in the SynZIP library that we could use to similarly enhance the α fragment binding.

Reviewer #2:

1. *In Fig. 2c-e, I would suggest to show the case of the original full T7 (T7*) RNAP for comparison of the transcriptional activity. This is done in Fig. 3b, but it would be useful here as well.*

The toxicity of the full length T7* RNAP control makes it difficult to compare directly to the split T7 systems. This toxicity causes the fluorescence to artificially appear low at the same expression levels shown in Figure 2, so we have not included the data.

2. *Does any of the systems constructed here affect the host cell, e.g its growth rate? Does this impose a limitation to the expression of for example the core fragment?*

We have added a section to the Supplementary Information (Supplementary Information Section III.F.) where we measure the growth impact of the fragments and combinations of fragments.

3. *The core fragment appears to be always limiting in the cases studied here. While this is certainly desirable from the application point of view, it would be a nice control supporting the overall picture to show a case, where the core fragment is not limiting, for example in the competition assay (Fig. 4) or for the repressor systems (sigma_null, fig. 5a,c). In the latter case, this control would also provide clearer evidence that repression indeed functions by sequestering the core part and not by some other mechanism. At least a good argument in support of the mechanism should be given.*

Any of the σ fragments can repress the activity of the others through competition for the core fragment, and the null fragment likely retains this ability as it has only one AA changed from σ_{CGG} . The curve showing repression of σ_{T7} using the null fragment (Figure 5B) is remarkably similar to the results obtained for σ_{T3} competing with σ_{K1F} (Figure 4A-B) and is consistent with the model for σ fragment competition (Supplementary Section IV.B, Equations 24, 30), suggesting that the repression effect is simply competition for core. We have clarified this point in the appropriate section of the Results.

4. *I am not sure that the term 'resource allocator' as used in fig. 1 (for the part of the system encoding the core fragment) is appropriate. This part only sets the overall level of the limiting resource, but allocation to different actuators, the eventual task, is a collective property of the whole system including the sigma-fragment encoding parts. I would thus rather call the whole system a resource allocator and refer to the core-encoding subsystem as the limiting resource.*

We refer to the resource allocator as a separate genetic system (e.g., plasmid) that sets the total availability of the resource. In this case, this is the third plasmid shown in Figure 1A. Thus, we have maintained the “resource allocator” descriptor for this plasmid.

5. *Overall, I think Fig. 1 could be improved. It is a bit confusing that fig. 1b and c are actually only very briefly discussed much later in the paper, maybe this could be moved to the discussion section (with a separate figure) or to the supplement.*

We have edited the Introduction to better describe these subfigures. For us, this is the key motivation behind the work. We have also added significantly more analysis regarding the mathematical model and the theoretical linkage between Figures 1 and 4. The detailed derivations and analysis of the model are provided in Supplementary Information Section IV.

Acceptance letter

24 June 2014

Thank you again for sending us your revised manuscript. We are now satisfied with the modifications made and I am pleased to inform you that your paper has been accepted for publication.

Thank you very much for submitting your work to Molecular Systems Biology.

Reviewer #2:

The revision of the paper clarifies several minor issues with this paper, which (as already said in the previous report) describes beautiful innovative work. In particular, I think the revision of the discussion of fig. 1 (motivation of the work) and the new data on growth impact of the system should be quite useful. I recommend publication of the manuscript in its present form.